# Food Protein-Derived Antioxidant Peptides: Molecular Mechanism, Stability and Bioavailability

**DOI:** 10.3390/biom12111622

**Published:** 2022-11-03

**Authors:** Yongsheng Zhu, Fei Lao, Xin Pan, Jihong Wu

**Affiliations:** College of Food Science and Nutritional Engineering, China Agricultural University, Beijing 100083, China

**Keywords:** bioactive peptides, food sources, antioxidant activity, molecular mechanism, stability and bioavailability, food applications

## Abstract

The antioxidant activity of protein-derived peptides was one of the first to be revealed among the more than 50 known peptide bioactivities to date. The exploitation value associated with food-derived antioxidant peptides is mainly attributed to their natural properties and effectiveness as food preservatives and in disease prevention, management, and treatment. An increasing number of antioxidant active peptides have been identified from a variety of renewable sources, including terrestrial and aquatic organisms and their processing by-products. This has important implications for alleviating population pressure, avoiding environmental problems, and promoting a sustainable shift in consumption. To identify such opportunities, we conducted a systematic literature review of recent research advances in food-derived antioxidant peptides, with particular reference to their biological effects, mechanisms, digestive stability, and bioaccessibility. In this review, 515 potentially relevant papers were identified from a preliminary search of the academic databases PubMed, Google Scholar, and Scopus. After removing non-thematic articles, articles without full text, and other quality-related factors, 52 review articles and 122 full research papers remained for analysis and reference. The findings highlighted chemical and biological evidence for a wide range of edible species as a source of precursor proteins for antioxidant-active peptides. Food-derived antioxidant peptides reduce the production of reactive oxygen species, besides activating endogenous antioxidant defense systems in cellular and animal models. The intestinal absorption and metabolism of such peptides were elucidated by using cellular models. Protein hydrolysates (peptides) are promising ingredients with enhanced nutritional, functional, and organoleptic properties of foods, not only as a natural alternative to synthetic antioxidants.

## 1. Introduction

Rapid and uncontrolled urbanization, the globalization of unhealthy lifestyles, and environmental and climatic degradation resulting from human development activities have contributed to the high prevalence of non-communicable chronic diseases (NCDs) worldwide. According to statistics, premature deaths due to NCDs exceed 41 million per year, equivalent to 71% of total global deaths [1]. There is growing evidence that oxidative stress caused by the disturbance of redox homeostasis in living organisms is involved in the pathogenesis and development of many chronic diseases, such as cancer, atherosclerosis, and diabetes [2,3]. Reactive oxygen species (ROS) are a class of free radical species produced mainly by the mitochondrial respiratory chain and are involved in oxidative stress signalling in normal cells. However, if the accumulation of ROS exceeds the capacity of the cellular free radical scavenging system, these reactive species trigger uncontrolled reactions with non-target biomolecules (lipids, proteins, and DNA) and cells, and mediate the subsequent activation of pro-inflammatory or pro-apoptotic pathways. This situation requires additional supplementation to regulate the balance of antioxidants and oxidants in biological tissues. Since the beginning of this century, the World Health Organization has been calling for an increase in the consumption of dietary sources of antioxidants worldwide, as food is a natural and sustainable source of these compounds [4]. The application of antioxidant active peptides in the prevention and management of oxidative damage and related pathologies in the body has been extensively studied over the past decades. As of June 2022, 772 peptide sequences with biological antioxidant functions were registered in the BIOPEP database, second only to angiotensin-converting enzyme-inhibiting peptides, reflecting their great commercial exploitation value. The sources of these active peptides cover a wide range of human edible biological resources on earth, including animals, plants, and algae. They can be produced from low economic-value catches or crops, or various edible or non-edible by-products of food processing, even involving some refined products. Most biofunctional peptides are produced mainly through enzymatic hydrolysis of proteins, either in vivo during gastrointestinal digestion, controlled degradation using appropriate exogenous proteases, or during specific food processing (e.g., ham and milk fermentation) [5] (Figure 1). Traditionally, the characterization of peptides follows a standardized procedure, which simply includes the selection of the original protein, enzymatic hydrolysis, isolation, purification, and identification, and after the last step, information on the activity, amino acid sequence, structure and corresponding functional properties of the candidate peptide can be largely determined [6]. However, this approach is expensive and time-consuming, and more importantly, it does not meet the requirements of industrial scale-up production. In recent years, the establishment of emerging bioinformatics analysis systems (in silico) has provided a new possibility for the study of biopeptides including antioxidant peptides. In addition, besides their potential as therapeutic agents, the research value of antioxidant peptides is also reflected in their applications as food additives, nutritional fortification in health foods, and anti-aging and photoprotective components in cosmetics [7,8]. Numerous experiments have shown that the addition of food-based antioxidant protein hydrolysates or peptides as antioxidants can effectively inhibit lipid peroxidation during food transportation and storage, thus maintaining the stability of food flavour and nutritional quality (vitamins and essential unsaturated fatty acids) [9,10]. Therefore, the development of natural antioxidant peptides from food or other readily available sources as alternative food preservatives may help to alleviate consumer concerns about the potential toxicity risks associated with the widespread use of synthetic antioxidants in current food formulations.

In this review, we highlight recent advances in the discovery of antioxidant peptides from edible sea-based and land-based plant and animal sources and food-processing by-products, particularly those discovered in the past five years. In addition, we present a detailed review of the latest knowledge on several key issues in the study of such biopeptides, namely molecular mechanisms, digestive stability, and bioavailability, and highlight their potential role as food additives and functional ingredients. Finally, the main challenges for the commercialization of antioxidant peptides and the future directions of research are also pointed out.

## 2. Methods of Activity Evaluation and Mechanisms of Action of Antioxidant Peptides

### 2.1. Chemical Evaluation Methods and Activity Mechanisms

#### 2.1.1. Free Radical Scavenging

Depending on the mechanism of free radical deactivation, antioxidant peptide molecules can achieve quenching of reactive oxygen/nitrogen species through two pathways, namely, hydrogen atom transfer (HAT) and single electron transfer (SET) reactions [11]. Specifically, antioxidant capacity evaluation methods such as oxygen radical absorbance capacity (ORAC) and total radical capture antioxidant parameter (TRAP) assays using HAT as the reaction principle to measure the ability of antioxidants to achieve free radical scavenging through proton donation [12,13]. The bond dissociation energy and ionization potential (IP) of the hydrogen atom donor group of the antioxidant are key parameters to measure the reaction strength of the HAT reaction. In contrast, SET-based methods such as DPPH radical (or DPPH^•^) scavenging capacity, ferric reducing antioxidant capacity (FRAP), and ABTS radical (or ABTS^•+^) scavenging capacity reflect the ability of an antioxidant to reduce a radical, metal, or carbonyls by losing an electron, and their relative reactivity is influenced by the deprotonation of reactive functional groups and IP [14]. However, it has been shown that HAT and SET mechanisms occur together in almost all samples, the dominant mechanism being determined by the influence of the structure of the antioxidant on its systemic solubility and partition coefficient [15]. Apart from the above comprehensive evaluation methods, the antioxidant capacity of peptides can also be investigated by the scavenging efficiency of other free radicals such as superoxide anion (O_2_^•-^) and hydroxyl radical (^•^OH). O_2_^•-^ is the reduced form of molecular oxygen with a weak oxidant; however, it can be decomposed to form more potent and reactive ROS such as ^•^OH, thus causing potential harm to the organism. It was shown that superoxide radical scavenging assays were positively correlated with ABTS, DPPH, and FRAP, whereas the quenching mechanism of ^•^OH was more related to the hydrogen atom transfer ability of antioxidants [14,15,16].

#### 2.1.2. Chelation of Metal Ions

It is well known that metals play an important role in various physiological activities of biological systems. Disruption of metal ion homeostasis may lead to the uncontrolled metal-mediated formation of harmful free radicals, participate in DNA base modification, enhance lipid peroxidation, and alter calcium and thiol homeostasis [17]. As an example, the toxic effect of ferrous (Fe^2+^) overload in human plasma has been shown to catalyse the production of destructive ^•^OH via the Fenton reaction. Similar to iron, copper plays a catalytic role in the formation of ROS and catalyses the peroxidation of membrane lipids [17,18]. In addition, when organisms are exposed to redox inert metals such as cadmium (Cd), arsenic (As), and lead (Pb), they can show their toxic effects by binding to protein sulfhydryl groups and depleting glutathione, an endogenous regulatory metabolite with antioxidant and integrative detoxifying effects [18,19]. Under normal conditions, the body can maintain stable intracellular metal levels through effective sequestration and translocation of overloaded metals by various regulatory proteins and peptide detoxifiers. However, when this burden exceeds the autoregulatory capacity, the supplementation of exogenous chelators becomes particularly important. Previous studies have shown that the chelation of redox-active metals such as iron (Fe), copper (Cu), chromium (Cr), and cobalt (Co) by antioxidant peptides is one of the main ways to convey their activity. Specifically, this complexation is manifested as an electrostatic attraction between the electron-donating group and the electron acceptor, i.e., the metal ion, of a peptide with multiple available coordination sites. Taking iron redox couple as an example, antioxidant peptides can prevent their toxic effects by (1) chelating ferrous ions to prevent the reaction with molecular oxygen or peroxides; and (2) chelating iron to maintain its original valence so that it cannot reduce molecular oxygen [20,21]. The metal ion chelating activity of antioxidant peptides is generally calculated by their ability to chelate Fe^2+^ or Cu^2+^.

#### 2.1.3. Lipid Peroxidation Inhibition (LPI)

Lipid oxidation in food systems can lead to undesirable flavors and the formation of toxic compounds. In biological systems, however, a clear link has been established between lipid oxidation products and the etiology of many diseases such as atherosclerosis, Alzheimer’s disease, and cancer. As mentioned earlier, lipid peroxidation reactions can result from direct oxidation by reactive free radicals or can be triggered by the mediation of redox-active metals. Transition metals such as Fe^2+^ and Cu^2+^ are pro-oxidants that catalyze the breakdown of hydroperoxides into free radicals, thus indirectly initiating the oxidative degradation of lipids [17]. Thus, like other antioxidants, antioxidant peptides can act against antioxidants in two established ways, namely by protecting target lipids from oxidative initiators or by impeding the propagation of chain lipid peroxidation. In the first case, the active peptide inhibits the production of ROS or scavenges the active species that cause oxidative initiation such as O_2_^•-^. In the second case, the antioxidant peptide molecule can intercept lipid peroxyl radicals (LOO^•^) generated by lipid autoxidation by providing a hydrogen atom, forming a less reactive hydroperoxide, thus interrupting the chain reaction of lipid radicals [22,23]. In addition, it is worth mentioning that the strong emulsifying properties of these antioxidants may give them a unique inhibitory capacity, since this allows them to adsorb well on the surface of the lipid molecules, blocking their contact with oxygen. For example, García-Moreno et al. [24] obtained emulsifier peptides from potato test streams and investigated their antioxidant activity by combining bioinformatics and top-down proteomics approaches. The results showed that the peptides containing the FCLKVGV sequence resulted in highly oxidatively stable fish oil-water emulsions and showed good DPPH antioxidant activity. In another study, carp caviar protein hydrolysates with free radical scavenging and chelating properties were also reported to be effective in delaying the loss of tocopherols and polyunsaturated fatty acids in cod liver oil-water emulsions [25]. Lipid peroxidation assays are usually based on β-carotene/linoleic acid emulsion systems. In addition, ferric thiocyanate, peroxide, and thiobarbituric acid (TBARS) assays can also be used for the investigation of lipid peroxidation inhibitory activity of antioxidant peptides.

In conclusion, chemical evaluation of the antioxidant capacity of peptides is essential for the understanding of their molecular mechanisms and the subsequent biological work. However, due to the lack of standardized methods, there is still no single evaluation tool to describe the overall antioxidant capacity of antioxidant peptides and to support the comparison between different studies, which has caused a lot of problems for the advancement of research related to antioxidant active peptides. Here, the continuous improvement of bioinformatics tools may provide a simple, rapid, and low-cost solution for the validation and comparison of peptide chemical activities.

### 2.2. Biological Evaluation Methods and Activity Mechanisms

#### 2.2.1. In Vitro Cell Aspects

The cellular antioxidant assay is the classic tool used to evaluate or screen for biologically effective active ingredients before in vivo testing. This is not difficult to understand, partly because cell membranes are lipid- and protein-based and therefore susceptible to oxidation. More importantly, the cytoprotective effect of antioxidant active peptides on damaged cells can more directly reflect the oxidation-induced tension of the organism. A series of human and animal cultured cells have been used to construct oxidative stress models to study the ability of peptides to scavenge ROS, inhibit oxidative damage, and maintain cellular redox homeostasis. The isolated cell lines involved here include human umbilical vein endothelial cells (HUVECs), neuroblastoma cells (SH-SY5Y), intestinal cancer cells (Caco-2), cervical cancer cells (Hela), and hepatocellular carcinoma cells (HepG2); as well as mouse macrophages (RAW264.7), Saccharomyces cerevisiae BY4741, and rat hepatocytes and pheochromocytoma cells (PC12) [26]. H_2_O_2_ is the most commonly used inducer of cellular stress formation, characterized by its ability to cross cell membranes in a free-diffusion manner and then convert to highly reactive ^•^OH by the Fenton reaction to stimulate the cellular stress response, and the accumulation of ^•^OH can activate the apoptotic program of cells [26,27]. Besides, LPS (lipopolysaccharide), AAPH (2,2′-azobis(2-methylpropionamidine) dihydrochloride), and TbOOH (tert-butyl hydroperoxide) can also be used to evaluate the preparation of the model. At the biological level, dynamic changes in stress levels and the antioxidant capacity of functional peptides can be assessed by monitoring the release of biomarkers of oxidative damage. These biomarkers, such as malondialdehyde (MDA), carbonyl, and 8-hydroxydeoxyguanosine (8-OHdG) originate from oxidative deterioration of biomolecules, i.e., lipids, proteins, and DNA [20]. On the other hand, cellular antioxidant activity (CAA), an assay to quantitatively investigate the ability of active compounds to quench peroxyl radicals induced by AAPH in HepG2 cells, has also been widely used for in vitro biological analysis and activity comparison of antioxidant peptides, as the results are usually expressed as micromoles of quercetin equivalent (QE) per 100 g of peptide [28]. From this perspective, CAA may play an important reference value in building a database of antioxidative biopeptides.

It is generally accepted that antioxidant molecules have direct and indirect antioxidant capacities at the cellular and organismal levels, depending on their mechanisms. Direct antioxidant capacity is manifested by the scavenging of reactive oxygen and nitrogen species by antioxidants through depletion or chemical modifications. In contrast, the pathway by which antioxidants protect against oxidative damage by upregulating the activity and expression of endogenous antioxidant enzymes and non-enzymes is considered an indirect antioxidant capacity. Numerous studies have shown that when the balance between intracellular oxidant production and antioxidant concentration is disrupted, exogenous antioxidant peptides can act together through both pathways, exhibiting efficient inter-biomolecular synergism, hence such antioxidants are also referred to as bifunctional antioxidants [29,30,31]. The Keap1/Nrf2/ARE (full name Kelch-like ECH-associated protein 1-nuclear factor erythroid 2-associated factor-antioxidant response element) system is a key transcription factor regulatory pathway that coordinates the expression of cellular antioxidant protective genes [32] (Figure 2). Nrf2 is a transcription factor that primarily regulates the cellular response to oxidative stress. Under normal physiological conditions, intracellular levels of Nrf2 are low because Keap1 (a cytoplasmic protein) can sequester it in the cell membrane, promoting its ubiquitination and proteasomal degradation. When cells are exposed to oxidative stress, ARE (a DNA sequence responsible for regulating cellular antioxidant and cytoprotective responses) induces the decoupling of Keap1 from Nrf2 ubiquitination. Activated Nrf2 allows for nuclear translocation, where small muscle neurofibrosarcoma proteins (sMAF) will form a heterodimer with it, and this heterodimer subsequently acts on the ARE to activate the expression of a large number of genes encoding cellular antioxidant enzymes and proteins [33,34,35,36,37,38]. Among the endogenous enzymes and non-enzymatic systems, superoxide dismutase (SOD), catalase (CAT), glutathione peroxidase (GPx) and glutathione (GSH) constitute the regulatory system of oxidative stress. Peptide-mediated detoxification properties can also be investigated by measuring the activity and levels of these markers. As an example, incubation with the tripeptide PHP (an antioxidant peptide obtained from Chinese liquor) reversed an AAPH-induced decrease in CAT, SOD, and GPx activities and ROS, MDA, and GSSG (oxidized glutathione) production in HepG2 cells, effectively preventing oxidative damage to the cells [39]. This work also showed that PHP activated the signaling of cellular antioxidant defense by inducing dissociation of the Keap1/Nrf2 complex. This was corroborated by the results of the group of Tonolo et al. [40], which clarified that the antioxidant mechanism of the milk-derived peptide KVLPVEK is due to its binding to the Nrf2 site in the Keap1 pocket, which inhibits the interaction between Keap1 and Nrf2 and subsequently triggers the subsequent activation of the Nrf2 signaling pathway. In addition, some recent studies have led to the belief that inhibition of apoptosis by antioxidant peptides through regulation of the expression of related proteins is one of the molecular mechanisms. It is known that the accumulation of ROS can activate the apoptotic pathway by altering the mitochondrial membrane potential [30]. It was reported that pretreatment with the pine mushroom-derived peptides SDLKHFPF and SDIKHFPF could reverse the expression of anti-apoptotic protein Bcl-2 and pro-apoptotic protein Bax in LPS-induced RAW264.7 macrophages and effectively improve ROS-induced mitochondrial dysfunction [41]. In vitro biological analysis of rapeseed-derived peptide, WDHHAPQLR, revealed a consistent antioxidant molecular mechanism to inhibit apoptosis [42]. H_2_O_2_-induced HUVECs cells were incubated with WDHHAPQLR (0.2 mM), and the fully expressed Bcl-2 could increase the concentration of intracellular GSH and other reducing agents, which was conducive to the correction of the dysregulated redox homeostasis in cells.

However, in vitro cell experiments have their limitations, such as expensive detection reagents, easy contamination of the cell culture process, and not reflecting the real situation in vivo. In addition, it is worth noting that the dose of in vitro samples is usually much higher than the actual utilization level of the organism. Therefore, further in vivo evaluation of the effects is necessary.

#### 2.2.2. In Vivo Animal Aspects

A variety of model organisms can be used to carry out in vivo studies on the effects, activity mechanisms, and bioavailability of dietary antioxidant peptides. Among them, mice and rats are the most commonly used mammalian models. Some non-mammalian models, such as Drosophila, Caenorhabditis elegans, zebrafish, and turbot, have also been developed in recent years as ideal models for the biological analysis of pharmacologically active compounds due to their short life cycle, ease of cultivation, and stress response consistent with that of humans [26,43,44]. For example, a recent study investigated the physiological functions of two novel antioxidant peptides, D2-G1S-1 (VENAACTTNEECCEKK) and G2-G1S-2 (VEGGAACTTGGEEGCCEKK), using the Caenorhabditis elegans model [45]. After pretreatment with D2-G1S-1 or G2-G1S-2 (1.0 mM) for five days, a significant increase in nematode resistance to oxidative stress induced by paraquat (treated at 100 mM for 6 h) was observed. In the same animal model, the delipidated round trehalin-derived peptide ILGATIDNSK was also identified as an antioxidant peptide with protection against oxidative stress [43]. Two novel peptides isolated from yak collagen (GASGPMGPR and GLPGPM), were also reported to have similar in vivo antioxidant defense capacity [46]. Another study, based on a zebrafish model, investigated the protective effect of the antler protein-extracted peptide TAVL against AAPH-induced oxidative stress [47]. A dose-dependent reduction in apoptosis, ROS production, and lipid peroxidation levels was observed in peptide-incubated zebrafish embryos. The above study highlights the potential of several natural antioxidant peptides as functional substances for the prevention of oxidant damage in the organism. In addition, to investigate the therapeutic efficacy of antioxidant peptides on postmenopausal osteoporosis in organisms, Mada et al. [48] group conducted an interesting experiment. They constructed an ovariectomized (OVX) osteoporosis rat model and then analyzed the physiological function of the novel peptide VLPVPQK (derived from buffalo milk casein) by continuous gavage treatment. It was observed that the heptapeptide partially reversed the decrease of SOD and CAT activities, the decrease of GSH level, and the increase of MDA level in the serum of rats induced by OVX. In addition, the results also revealed that the antioxidant peptide could alleviate bone loss in OVX rats by enhancing the stress defense capacity. In another study, the alleviation of ovarian oxidative stress levels by tilapia skin peptide (TSP) was shown to improve primary ovarian failure in mice [49]. These studies suggest that the development of antioxidant peptides as pharmaceutical components may have efficacy in alleviating or treating multiple pathologies.

Compared to in vitro cellular assays, kinetic assays provide a more realistic picture of the antioxidant capacity of biopeptides under normal physiological conditions. In addition, some beneficial functions that could not be explored by chemical and cellular modeling approaches, such as disease prevention and treatment, life extension, anti-fatigue, and memory improvement, have been rapidly developed and refined. These insights are critical for the design and guidance of later clinical trials. However, animal studies and human trials are costly and complex and have long testing cycles. Therefore, the development of a simple, rapid, low-cost, high-throughput, and comprehensive antioxidant evaluation system is important for the commercialization of biofunctional peptides.

## 3. Sources of Antioxidant Peptides

### 3.1. Marine Sources

As the world’s population grows and the 2030 Sustainable Development Goals are advanced, the global fisheries and aquaculture industry is booming in terms of size and consumer markets. With a record total production of 214 million tonnes and a per capita consumption of 20.2 kg in 2020, the industry will play an increasingly important role in providing food and nutrition for the future [50]. The booming fishery industry is driving the annual growth of fish protein as a percentage of the total protein consumption of the global population, and the rapid development of the fish protein hydrolysate market. The global fish protein hydrolysate market was expected to grow at a compound growth rate of over 4.5%, with a total industry value of over USD 475 million by 2026 [51]. Considering the unparalleled biodiversity of the vast ocean and the unique adaptability of these organisms to adverse environmental conditions such as darkness, cold, and high pressure, many species of aqueous feedstocks such as molluscs, crustaceans, as well as algae, and other aquatic plants, including fish, have been reported as high-quality precursor protein sources for many bioactive peptides such as antioxidant peptides [52,53,54]. For example, previous studies have successfully extracted antioxidant active peptide fractions from Pinctada fucata [55], Ribbon jellyfish [56], shrimp [57], oyster [58], and Palmaria palmata proteins [59].

The processing of bioactive peptides is not limited to the edible part of fishery products, in other words, the alternative use of fishery by-products to produce these high value-added products has become a good initiative to improve the environment and generate profit and income. In the case of fish, for example, with the development of the fish processing industry, the amount of discarded fish and fish by-products is increasing, and these waste products include heads, guts, bones, skins, and scales, which may reach up to 70% of the total weight of the fish [60]. In the past, fish by-products were either discarded as waste, used directly as feed for aquaculture, livestock, pets, or fur-bearing animals, or used as fish silage and fertilizer. Based on advances in processing technology and the potential for bioactive ingredient development, the work of researchers in the food, nutritional, pharmaceutical, cosmetic, and medical sectors have shifted more towards the use of these wastes in recent years. Table 1 lists recent studies of antioxidant protein hydrolysates/peptides found in various parts of marine organisms. It is well known that by-products of fish processing are an excellent source of collagen (or gelatine), especially in fish bones and skin. In skipjack tuna, for example, the organic component of fish bones, which accounts for 30% of the material, is composed of collagen [61]. For this reason, Yang et al. [62] group investigated the sequence of antioxidant active peptides of bonito bone gelatine. Five amino acid sequences, GPDGR, GADIVA, GAPGPQMV, AGPK, and GAEGFIF, were screened. Among them, the half-effective inhibitory concentrations (EC_50_) of GADIVA and GAEGFIF against DPPH radicals, hydroxyl radicals, superoxide anion radicals, and ABTS radicals were 0.57 and 0.30 mg/mL, 0.25 and 0.32 mg/mL, 0.52 and 0.48 mg/mL and 0.41 and 0.21 mg/mL, respectively, indicating that they have strong chemical antioxidant capacity. Recent work highlighted the cytoprotective mechanism of grass carp scale gelatine hydrolysate (GSGH) on oxidatively damaged cells [63]. It was noted that GSGH pretreatment could increase SOD, CAT, and GPx activities, reduce ROS and MDA content, as well as attenuate cell membrane and DNA damage in HepG2 cells, thereby alleviating H_2_O_2_-induced cell damage. Jellyfish proteins are another highly used precursor protein for antioxidant peptide development and production, as their by-product gonad contains approximately 50% of the protein [64]. To date, multiple peptide fragments with chemical and/or biological antioxidant activity such as AAGPAGPDGR, GCGLGDPPGHGK, WGPGPPGDLGAA, and SY have been identified from jellyfish proteins [56,64]. Among them, the dipeptide SY showed high scavenging efficiency against the three radicals DPPH^•^, ^•^OH, and O_2_^•-^, as evidenced by the half-inhibitory concentration (IC_50_) values of 84.623 µM, 1177.632 µM, and 456.662 µM, respectively. Also, the excellent ACE inhibitory activity of SY was observed (IC_50_ 1164.179 µM). This observation may verify the conclusion of Zheng et al. [65] that the existence of antioxidant amino acid residues with electron/hydrogen donor capacity, such as Tyr, is the dominant factor for the higher radical scavenging activity of dipeptides. In addition, the use of all parts of low-value pelagic fish for the processing of bioactive protein hydrolysates or peptide products has recently emerged in industries such as food and pharmaceuticals. For example, the global annual catch of Round scad exceeds 5 million tons, ranking third among all single species in the world [43]. However, due to its relatively small size, dark colour, susceptibility to spoilage, and poor taste, it is not suitable for commercial fish consumption [66]. In this regard, the biological antioxidant capacity and mechanism of Round scad protein-derived peptides WCPFSRSF and ILGATIDNSK have been confirmed [43,67]. Thus, a broad range of marine organisms or by-products can be exploited as raw materials for the elaboration of natural antioxidants, meaning that it has the potential to be used as a functional ingredient in the food, cosmetic, and pharmaceutical industries.

### 3.2. Dairy Sources

The health benefits of milk have been known since ancient times. In particular, in addition to its high value of nutrients, milk also contains antioxidants. Dairy products such as milk, yogurt, fermented milk, and cheese have been found to have antioxidant effects, possibly due to the functional activity of their protein components (casein and whey proteins) and/or peptide fragments from different protein fractions in the matrix [68,69]. Recent investigations have identified several potentially bioactive peptides from the protein fractions of mammalian milk and other dairy products that were evaluated for their free radical scavenging activity (Table 1). Over the past decade, camel milk-derived bioactive peptides have begun to attract significant interest from many researchers and have been evaluated for antioxidant, antihypertensive, antidiabetic properties, and antimicrobial properties. For example, work by the group of Zhang et al. [70] analyzed the content of endogenous bioactive peptides in dromedary and bactrian camel milk and the motifs of these peptides. The peptide sequencer showed that 14.6% and 15.7% of the quantified peptides from dromedary and bactrian camels were biologically active, with dipeptidyl peptidase IV inhibitors (39.93%) predominating, followed by ACE inhibitors (34.85%) and antioxidant activity (8.69%). Ibrahim et al. [71] further characterized the amino acid sequence of fragment peptides in the antioxidant active fraction of camel milk protein hydrolysate. As a result, 14 and 8 antioxidant peptides were recorded from casein and whey protein digests, respectively, and their activity mechanisms were associated with superoxide anion radical and DPPH radical scavenging capacity. In another study, eleven novel peptides (LLILTC, AVALARPK, YPLR, LSSHPYLEQLYR, TQDK, LAVP, NEPTE, VSSTTEQK, LAVPIN, KPVAIR, and LLNEK), identified from a Lactobacillus plantarum fermented camel milk, were reported to be effective in the scavenging activity of ABTS radical, hydroxyl radical, and superoxide radicals [72]. Similarly, two peptide sequences from camel milk protein, NEDNHPGALGEPV and KVLPVPQQMVPYPRQ, showed antioxidant activities against DPPH^•^ (IC_50_ 0.04 and 0.02 mg/mL), ^•^OH (IC_50_ 0.05 and 0.05 mg/mL), ABTS^•+^ (IC_50_ 0.1 and 0.01 mg/mL) and O_2_^•-^ (IC_50_ 0.045 and 0.3 mg/mL) [73]. Besides, the results highlight the excellent and sustained inhibition of peroxidation of linoleic acid emulsion by both peptides during storage (60 °C for seven days). Moreover, cellular experiments revealed that the peptide KVLPVPQQMVPYPRQ was able to inhibit the proliferation of cancer cells HepG2 by increasing the endogenous antioxidant defense through the upregulation of the mRNA expression level of SOD. In terms of protein composition, camel milk has similar β-casein content as human milk and does not have the allergenic milk protein β-lactoglobulin [74,75]. For these reasons, camel milk is an exciting and suitable product both as a future alternative to cow’s milk-based infant formula and as a material for the production of biopeptides with antioxidant activity.

Apart from camel milk-derived peptides, previous studies have investigated some potential bioactive antioxidant peptides in the protein fractions of dairy products of bovine and sheep milk origin. Chhurpi is a traditional cheese product made from the fermentation of lactic acid bacteria (LAB) from cow’s milk, and a recent study evaluated the antioxidant potential of the enzymatic hydrolysis products of this cheese product [76]. Among the Chhurpi products prepared with different fermentation starters, the highest antioxidant activity was observed for cheese produced with *Lb. delbrueckii* WS4 after digestion with a combination of pepsin and trypsin, exhibiting DPPH^•^ scavenging activity (0.370 ± 0.005 mg AAE (ascorbic acid)/g sample), O_2_^•-^ scavenging activity (2.214 ± 0.023 mg AAE/g sample), reducing power (1.122 ± 0.009 mg AAE/g sample) and total antioxidant activity (1.959 ± 0.023 mg AAE/g sample). A similar work evaluated the antioxidant properties of water-soluble peptides from fresh buffalo cheese [77]. The results highlighted the outstanding quenching ability of a 20 mg/mL mixture of water-soluble peptides against ABTS radical (63.27 ± 0.18%) and DPPH radical (78 ± 0.38–80 ± 0.15%). During milk fermentation, LAB strains have been reported to release antioxidant peptides such as VAPFPEVFGK, LLVYPFPGPLH, and FVAPEFVGKEK [78]. Differences in the peptide profiles of cheese products emerged due to the different specificity of the initiator proteases in hydrolyzing milk proteins. These results confirm the superior suitability of cheese-derived peptides as functional components for free radical scavenging in vitro, the actual efficacy of which needs to be further validated by cellular and oral administration.

Also of interest is the high sequence homology of milk proteins among different species (cows, goats, and sheep) [79]. For example, in silico coupled with in vitro analysis demonstrated that the comparison of amino acid sequences of the four casein α_s1_-, α_s2_-, β-, and κ-caseins of caprine and bovine milk showed that the similarity could reach 88%, 88%, 91%, and 85%, respectively [80]. The in silico proteolysis-based homology analysis of cow and yak milk casein between the same specie revealed a high degree of amino acid sequence identity for α_s1_-, α_s2_-, β-, and κ-casein (98.99%, 98.07%, 100%, and 97.11%, respectively) [81]. This implies that extensive studies on the peptide profile and bioactivity of a given mammalian milk casein can shorten the time required to screen for bioactive peptides present in different protein sources and could allow the discovery of new and sustainable precursors of known bioactive peptides.
biomolecules-12-01622-t001_Table 1Table 1Marine and diary sources of antioxidative hydrolysates and peptides.SourceExtraction Method(s)Extraction ToolHydrolysate Name/Peptide SequenceActivity Evaluation MethodsRef.Silver carp muscleEnzymatic and SGIDAlcalase + Pepsin and trypsin LVPVAVFISTSLPVMYPGIGDRADLVHVQChemical (ORAC, DPPH, FRAP, and LPI)in vitro cellular[82]Snakehead soupSGIDPepsin and trypsinPGMLGGSPPGLLG-GSPPSDGSNIHFPNChemical (DPPH, Fe^2+^ chelating, ^•^OH scavenging, and reducing power)in vitro cellular[83]SalmonChemicalsynthesis-PMRGGGGYHYPMRGGGYHYPMRGGYHYPMRGYHYPMRYHYYHYChemical (ORAC)in vitro cellular[84]Mackerel muscleEnzymaticProtamexALSTWTLQLGSTSF-ASPMLGTLLFIAIPIChemical (DPPH)in vitro cellular[85]Barred mackerel gelatineEnzymaticAlcalase and actinidinFraction 1Fraction 2Fraction 3Chemical (DPPH, FRAP, Fe^2+^ chelating, and ^•^OH and O_2_^•-^ scavenging)[86] Rainbow trout framesMicrowave pretreatment assisted (MPA) + enzymatic + SGID;MPA + SGIDAlcalase +Pepsin and trypsinSGID-MPCESGID-NPMEChemical (ABTS and FRAP)[87]JellyfishEnzymaticFlavourzymeJellyfish flavourzyme hydrolysateChemical (DPPH, ABTS, and FRAP)[88]Defatted round scadEnzymaticAlcalaseILGATIDNSKin vitro cellular[43]ShrimpEnzymaticAlcalaseMTTNIMTTNLChemical(DPPH and ^•^OH scavenging) in vitro cellular[89]Skipjack tuna boneSGIDPepsin and trypsinGPDGRGADIVAGAPGPQMVAGPKGAEGFIFChemical (DPPH, ABTS, and ^•^OH and O_2_^•-^ scavenging)[62]Milk caseinEnzymaticTrypsinLHSMKChemical (DPPH, FRAP, and ^•^OH and O_2_^•-^ scavenging) in vitro cellular[90]Fresh buffalo cheeseWater extractionUltrapure waterWater-solution peptidesChemical (DPPH and ABTS)[77]Bovine and caprine sodic caseinateEnzymaticSerine proteaseBovine caseinate hydrolysatesCaprine caseinate hydrolysatesChemical (ABTS and Cu^2+^ chelating)[91]Milk β-casein and κ-caseinChemicalsynthesis-ARHPHPHLSFMAVPYPQRNPYVPRKVLPVPEKin vitro cellular[40]Buffalo caseinChemicalsynthesis-VLPVPQKin vitro cellular[48]Fermented Rubing cheeseWater extractionDeionized waterYPFPGPIHChemical (DPPH)in vitro cellular[92]

### 3.3. Animal Sources

Despite growing concerns about the environment, dietary health, and animal welfare associated with meat production and consumption, data showed that global meat production continues to rise and is expected to reach 366 million tons by 2029, especially in developing countries [93]. Meat processing generates large amounts of animal waste and by-products unsuitable for human consumption, including bones, skin, feathers, blood, and offal, which can often account for more than half of the total weight of ketones [94]. These by-products are rich in nutrients such as proteins, fats, carbohydrates, and minerals. However, the underutilization of by-products is commonly reported in the meat industry. In recent years, advances in processing technology and improvements in waste recovery rates have not only created favorable conditions for the high-value utilization of these by-products but have also contributed positively to the reduction of environmental burden. Considering the importance of active proteins in human immune response, the development of proteins and their bioactive compounds from animal by-products has become an important strategy to seize the frontiers of life science and technology [95,96].

In this regard, biofunctional peptides have been identified from various livestock animal sources and processed meat products, including chicken, duck, goose, pig, sheep, and cattle, as well as fermented and dry-cured meat products. The biological activities of peptides cover a wide spectrum of actions such as antioxidant, antibacterial, anticancer, and antihypertensive [97]. Among them, antioxidant activity is a commonly found property of protein-derived peptides from various meat products and by-products. Recent reports of antioxidant protein hydrolysates/peptides obtained from edible parts of meat, waste or by-products, and processed meat products are shown in Table 2. Collagen is the most abundant protein among the various by-products obtained from the meat industry such as bone, cartilage, tendon, skin, and connective tissue, especially found in bovine bone, hides, and tendons [98]. A series of amino acid sequences with antioxidant activity have been identified from various collagen sources and validated in different oxidative systems [99,100]. Recently, Wang et al. [46] investigated the chemical and biological antioxidant effects of GASGPMGPR and GLPGPM, novel peptides derived from yak collagen. The activity mechanism of these two peptides was observed to be related to DPPH^•^, ABTS^•+^, ^•^OH, and O_2_^•-^ scavenging. Biologically, both GASGPMGPR and GLPGPM exhibited improved heat tolerance to heat-induced oxidative stress in Caenorhabditis elegans. The modulation of the antioxidant defense system of the model organism by GLPGPM was highlighted, as demonstrated by the pretreatment of 0.1 mg/mL of the peptide that increased the activity of SOD and CAT (33.18% and 101.80%, respectively) and reduced ROS and MDA accumulation (17.89% and 48.90%, respectively). Another work evaluated the antioxidant capacity of enzymatic hydrolysates of bovine bone collagen [101]. The results pointed out the DPPH^•^ (40.7%), ^•^OH (31.8%), and O_2_^•-^ (73.2%) scavenging activities of the hydrolysate (30 mg/mL). In addition, based on the good hygroscopic properties, the hydrolysate here is also considered to be added as a natural moisturizing ingredient in cosmetics. The multiple physiological activities of collagen peptides recovered from animal by-products, as well as their beneficial effects on bone, joint, and skin health, have been confirmed, and several drugs containing collagen hydrolysates have been introduced for joint injuries [99,102,103].

Moreover, more recent studies have focused on the discovery of antioxidant active peptides in fermented and dry-cured meat ready-to-eat products. The best-known intact antioxidants of meat origin are myostatin (β-alanyl-L-histidine) and anserine (N-β-alanyl-1-methyl-L-histidine). These are two endogenous antioxidant peptides, the concentration of the first peptide varies depending on the type of meat, while the latter is abundant in chicken meat. Their antioxidant activity is mainly related to the chelating activity of transition metals [104]. Additionally, the amino acid sequences of peptides that play a key antioxidant role in several regional types of ham have been identified; these are AEEEYPDL, SNAAC, GLAGA, and SAGNPN (Spanish dry-cured ham) [105,106,107], GKFNV and LPGGGHGDL (Chinese Jinhua ham) [108] and DLEE (Chinese Xuanwei ham) [109]. Also, three short peptides MWTD, APYMM, and FWIIE with ABTS^•+^ scavenging antioxidant capacity (IC_50_ 0.40, 0.12, and 0.23) were described as characteristic antioxidant peptides formed during the ripening of Chinese dry-cured mutton ham [110]. Interestingly, these peptides are produced naturally during ham maturation, which means that they are the result of proteolytic phenomena exerted by intramuscular peptidases [111]. Of course, the content and biological effects of active peptides in various types of dry-cured hams vary depending on the genetics of the ham used as raw material and the processing conditions and time [112,113]. For example, Wang et al. [114] quantified the number of endogenous peptides in Xuanwei ham, Jinhua ham, and lamb ham. Peptide composition analysis by UPLC-Q-TOF-MS/MS resulted in the identification of 346, 203, and 296 peptides from the three hams, respectively. Myosin, actin, myoglobin, troponin, and pyruvate kinase proteins were identified as the main reasons for the differences in peptide concentrations in the three dry-cured hams, which were essentially the result of genetic differences among the raw meat species.

### 3.4. Plant Sources

Plants, like animals, are an exemplary source of natural biopeptides, although the abundance of plant proteins in phytophagous crops or their agro-industrial by-products is relatively low. However, considering the advantages of high production volume, low unit cost, short yield cycle, freedom from regional religious or socio-cultural bias, and exquisite biological efficacy, phytogenic proteins are being more widely investigated as ideal precursor materials for biofunctional peptides. Antioxidant protein hydrolysates or peptides have been identified and characterized by numerous phytophagous crops and their by-products, such as legumes, grains, vegetables, fruits, seeds, husks, and leaves (Table 3). Given the global distribution of cultivation and protein content, the three major crops, rapeseed, cereals, and legumes, are the main phytochemical protein sources [115]. In this sense, this offers more potential opportunities for the availability, affordability, and diversity of bioactive peptides, including antioxidant peptides, as amply demonstrated by the qualitative and quantitative relationships between animal-derived biopeptides and their protein precursors [97]. However, numerous studies have shown that cereal and legume proteins do contain the most abundant peptide fragments with antioxidant activity among all investigated crop proteins. Among the major cereal cash crops, antioxidant peptide sequences have been identified in oats [116], rye [117], wheat [118], buckwheat [119], rice [2], corn [120], millet [121,122] and canary seeds [123]; meanwhile, among the legumes, soybean peptides have received the most attention, followed by chickpea, faba bean and mung bean [9,124,125,126].

Fruits and vegetables are known to be excellent sources of natural antioxidants such as phenols, flavonoids, alkaloids, saponins, and terpenoids with strong physiological activities. Recently, the biological activities of various antioxidant peptides from fruits and vegetables have attracted much attention from scientists. For instance, peptide fractions or amino acid sequences with organismal antioxidant protective capacity have been identified from edible parts or seeds of watermelon [127], sacha inchi [128], perilla [129], mulberry leaves [130], tricholoma matsutake singer [131], amaranth [132], and yam [133] by using enzymatic digestion or microbial fermentation. Of interest, mushrooms were reported to have the highest protein content among the 20 commonly consumed vegetable species [134]. Considering its considerable growth rate and impressive biological category, these undoubtedly provide an ideal material basis for the discovery of antioxidant mushroom peptides. In this regard, Agaricus bisporus is an extensively studied mushroom with a protein content of up to approximately 40% on a dry basis [134]. In a study reported on the antioxidant potential of A. bisporus protein hydrolysates, alcalase, pancreatin, and Flavourzyme were used individually or in combination for the preparation of hydrolysates [135]. It was found that alcalase hydrolysate and its 1–3 kDa fraction showed the strongest Fe^2+^ chelating ability (EC_50_ 0.96 and 1.2 mg/mL) among all hydrolysates and ultrafiltration fractions, respectively. It was also observed that the fractions with molecular weights of 1–3 kDa in the trypsin hydrolysate showed excellent DPPH radical scavenging activity (EC_50_ 0.13 mg/mL). The gastrointestinal enzyme hydrolysates of Agaricus bisporus and Terfezia claveryi proteins also exhibited antioxidant activity [136]. The hydrolysate of Agaricus bisporus protein exhibited excellent DPPH radical scavenging activity (73.68%), while the latter was more effective in inhibiting linoleic acid oxidation (85.85%) and Fe^2+^ chelation (21.36%). Another work performed a comparative analysis of the antioxidant activity of different fractions of King boletus mushroom protein hydrolysates [137]. The results highlighted the radical quenching ability of the fraction with a molecular weight below 1 kDa, i.e., this fraction (1 mg/mL) had the highest DPPH^•^ and ABTS^•+^ scavenging activities (12.08 and 94.14 mmol TE/mg protein, respectively). This result is consistent with previous reports on the <1 kDa fractions of corn gluten meal hydrolysate [138] and pea protein hydrolysate [139], probably because the shorter amino acid sequences of the low-molecular-weight peptides facilitate the exposure of reactive groups and thus enhance the accessibility of free radicals. Nevertheless, further experiments should be more concentrated on the bioactivity and mechanism to offer a theoretical basis for the digestive behaviour and the construction of the delivery system of mushroom peptides.

Furthermore, recent studies have shown that watermelon seeds can also be used as an alternative raw material for antioxidant-active peptide discovery because of their high protein content (30.63–43.60%) and balanced amino acid structure (Arg, Glu, Asp, and Leu) [140,141]. For example, Wen et al. [127] group identified five novel antioxidant peptides RDPEER, KELEER, DAAGRLQE, LDDGRL, and GFAGDDAPRA from watermelon seed protein hydrolysate using ultrasound pretreatment-assisted enzymatic assay. All these peptides were effective in improving the viability of H_2_O_2_-injured HepG2 cells, among which RDPEER was observed to have a prominent cytoprotective effect, as demonstrated by the increased CAT, SOD, and GPx activities and reduced ROS, Ca^2+^ and MDA levels in HepG2 cells at incubation doses of 12.5–100 µmol/L. In addition, these peptides were reported to have ideal DPPH^•^ clearance (IC_50_ 0.216 ± 0.01–0.435 ± 0.03), ABTS^•+^ clearance (IC_50_ 0.54 ± 0.02–1.23 ± 0.03), and ORAC (82.36 ± 1.2–130.67 ± 2.2 μM TE/mg). Another study by the same group highlighted the modulation of signaling pathways of biological antioxidant protection by watermelon seed protein hydrolysates [140]. They indicated that these antioxidants could activate the Nrf2/HO-1 (an inducible stress protein) pathway to enhance the resistance of RAW264.7 cells to H_2_O_2_-mediated oxidative damage. These results suggest that watermelon seed protein hydrolysate/peptide has potential as an alternative to synthetic antioxidants in food and as an active ingredient in drugs.

Besides, tomato seeds (GQVPP) [142], rapeseed (WDHHAPQLR) [42], peony seeds (SMRKPPG) [143], rhizomes (VTYM) [144], cherry (NLPLL) [145], spinach (YWTMWK) [146], almonds (LLPH) [147] and walnuts (PPKDW) [3] have also been reported to contain peptide antioxidants with chemical or biofunctional properties.
biomolecules-12-01622-t002_Table 2Table 2Animal sources of antioxidative hydrolysates and peptides.SourceExtraction Method(s)Extraction ToolHydrolysate Name/Peptide SequenceActivity Evaluation MethodsRef.Spanish dry-cured hamChemicalsynthesis-SNAACChemical (ORAC, ABTS, and LPI)[105]Porcine plasmaEnzymeticAlkaline proteaseWGPGVEChemical (DPPH, ABTS, Fe^2+^ chelating, and ^•^OH scavenging)in vitro cellular[148]Yak bones collagenChemical synthesis-GASGPMGPRGLPGPMChemical(ABTS, DPPH, and ^•^OH and O_2_^•-^ scavenging)in vitro cellular[46]Bovine bone collagenEnzymeticThermolysin-like Protease A69Bovine bone collagen hydrolysateChemical (DPPH, ^•^OH and O_2_^•-^ scavenging)[101]Dry-cured Xuanwei hamChemicalsynthesis-DLEEChemical (DPPH, ORAC, ABTS, and O_2_^•-^ scavenging)in vitro cellular[29]Pork sarcoplasmic and myofibrillar proteinMicrobial fermentationLactobacillus plantarum CD101, Staphylococcus simulans NJ201Sarcoplasmic protein hydrolysate;myofibrillar protein hydrolysateChemical (DPPH, ABTS, and Fe^2+^ chelating)[149]Chicken breastAcid extraction,SGID + acid extractionPorcine pepsin, trypsin, chymotrypsin, porcine pancreatic α-amylase, and porcine pancreatic lipase, HCl (0.01N)Cooked protein extracts;Cooked protein + SGID extractsChemical (ORAC, ABTS, DPPH, and FRAP)[150]Spanish dry-cured hamChemicalsynthesis-AEEEYPDLChemical (ORAC and ABTS)[107]Duck plasmaEnzymeticAlcalase LDGPTGVGTKEVGKRCLQLHDVKKLGAAGGVPAGChemical (DPPH, FRAP, ABTS, and Fe^2+^ Chelating)[151]Chinese dry-cured mutton hamSalt extractionPhosphateMWTDAPYMMFWIIEChemical (ABTS and LPI)in vitro cellular[110]Duck breastEnzymeticneutraseAGPSIVHLLCVAVFLLPHChemical (DPPH, FRAP, and ABTS)[152]Mutton ham, Xuanwei ham, Jinhua hamSalt extractionPhosphateMutton ham peptidesXuanwei ham peptidesJinhua ham peptidesChemical (DPPH, ABTS, Fe^2+^ chelating, and •OH and O_2_^•-^ scavenging)[114]
biomolecules-12-01622-t003_Table 3Table 3Plant sources of antioxidative hydrolysates and peptides.SourceExtraction Method(s)Extraction ToolHydrolysate Name/Peptide SequenceActivity Evaluation MethodsRef.Lotus seedEnzymeticFlavourzymeLotus seed protein hydrolysatesChemical (DPPH, FRAP, and H_2_O_2_ scavenging)[153]Wheat gluteninEnzymeticFlavourenzymeSavinaseSubsitilinSavinaseFlavourenzyme treated hydrolysatesSavinase treated hydrolysatesSubsitilin treated hydrolysatesAlcalase treated hydrolysatesChemical (FRAP and ABTS)[154]Soybean isolate proteinEnzymeticProtease from strain ERMR1:04Soybean proteinhydrolysatesChemical (FRAP, DPPH, ABTS and Fe^2+^ chelating)[155]Germinated amaranthSGIDPepsin and trypsinGAD 90GADW 90F1F2F3Chemical (ORAC)[132]WalnutChemicalsynthesis-PPKDWChemical (ABTS)[3]Corn gluten proteinEnzymeticAlcalaseAGLPMHALGAAGIPMHAIGAChemical (ORAC and ^•^OH scavenging)[156]Rhizome of white turmetic, turmeric and gingerSGIDPepsin and trypsinHVVVWTLTPLTPAVTYMRGPFHAEPPRGSGLVPKMSPVChemical (DPPH and ABTS)[144]Watermelon seedMPA + enzymeticAlcalaseRDPEERKELEEKDAAGRLQELDDGRLGFAGDDAPRAChemical (DPPH, ABTS, and ORAC)in vitro cellular[127]Pecan mealEnzymeticAlcalaseLAYLQYTDFETRChemical (DPPH, ABTS, and ^•^OH scavenging)[157]Tartary buckwheat albuminEnzymeticAlkaline proteaseGEVPWYMENFAFYRWChemical (DPPH and ^•^OH scavenging)[119]Peony seed dregEnzymeticAlcalasePeony seed dreg protein hydrolysatesChemical (DPPH, ABTS, Fe^2+^ chelating, and ^•^OH scavenging)[143]Sorghum kafirinEnzymeticPapainFraction 1Fraction 2Fraction 3Chemical (DPPH, ORAC, and Fe^2+^ chelating)[158]Finger millet seedsEnzymeticPepsinTrypsinTSSSLNMAVRGG-LTRSTTVGLGISMRSA-SVRChemical (DPPH)[122]Pine nut meal proteinEnzymeticAlcalaseKWFCTQWFCTChemical (FRAP, DPPH, and ABTS)in vitro cellular (CAA)[159]RapeseedChemical synthesis-WDHHAPQLRin vitro cellular[42]

## 4. Digestive Stability and Bioavailability of Antioxidant Peptides

As an ideal alternative to synthetic antioxidants at this stage, the stability and accessibility of functional protein hydrolysates or peptide derivatives in the complex and demanding digestive environment of the human body are undoubtedly decisive aspects in the biological validation of known and novel food-derived antioxidant peptides. However, the reality is that bioactive peptides, including antioxidant peptides, are still far from clinical application due to the lack of mature delivery and bioavailability support and the fact that the necessary biological analysis is still mostly at the in vitro cellular level. Like drug molecules and other functional components, in addition to their direct physiological effects in the intestinal wall, which effectively induce antioxidant defense mechanisms in the body, peptide molecules as therapeutic agents or health-promoting supplements must enter the portal circulation in their active form and exert systemic or local antioxidant effects.

To achieve this expectation, the bioactive peptides after oral administration need to be subjected to modification or degradation triggered by proteolytic enzymes in the gastrointestinal (GI) tract, while the peptide activity and function are also subjected to the possible impact of the toxic environment in the GI lumen such as potentially damaging secretions (including bile salts, acids and other digestive enzymes such as elastase) and various food-derived oxidants and toxins [87]. Peptides that survive gastrointestinal digestion or their released fragments must also overcome further hydrolysis by brush border peptidases and/or cell membrane peptidases of the intestinal membrane epithelium before they can be absorbed into the internal circulation by intestinal wall cells; there are four main mechanisms for the trans-cortical flux of peptides in this process as shown in Figure 3, including active transport by H^+^-coupled PepT1 and PepT2 transporters, Na^+^-coupled oligopeptide transport systems SOPT1 and SOPT2, passive bypass diffusion by intercellular tight junctions (TJs), and trans-cellular action in the form of endocytosis, depending on molecular size and structural properties such as hydrophobicity of peptides [126,160,161]. Finally, these cell-penetrating peptides also need to escape the breakdown of vascular endothelial tissue peptidases and soluble plasma peptidases, as well as the first-pass effect in the liver [162]. In fact, due to peptidase activity, most exogenous peptides have low stability and fast clearance in plasma [7]. In conclusion, in the face of the metabolic activity of peptidases in the gastrointestinal tract and plasma and the low permeability of the intestinal barrier, many therapeutic peptides have difficulty in maintaining their full activity or reaching the target site in very low amounts (1~2%) and are less likely to elicit a pharmacological response outside the GI tract [163]. Considering the strict ethical regulations of animal studies and the high cost and resource intensity of human trials, the evaluation, and integration of information on the digestive permeation behavior of bioactive molecules based on in vitro digestion and intestinal absorption models may provide valuable guidance for their in vivo effects and the subsequent development of co-delivery and bioavailability strategies.

To exert their biological activity, hydrolysates or peptides must be evaluated for digestibility and subsequent release of bioactive peptides in relevant in vitro intestinal models and the in vivo GI tract lumen. In vitro methods using cultures such as monolayers of human intestinal Caco-2 epithelial cells and in vivo models to determine permeability can aid in the prediction of oral bioavailability. The selective permeability of the intestinal barrier to candidate peptides is based on an understanding of the structural and chemical properties of the active compounds, the interactions in the gastrointestinal tract, and knowledge of the physiology of the GI tract [164]. As previously mentioned, it has long been known that hydrolysates with many short peptides, especially dipeptides and tripeptides, can lead to their better absorption and are more efficient than free amino acids and larger precursor peptide molecules [165]. If the MW of the molecule increases above 500 Da, the oral bioavailability decreases dramatically [162]. For example, the bioavailability of fractions of casein-derived peptides less than 500 Da was 16.23%, compared to 9.54% for fractions in the molecular weight range of 500–1000 Da [166]. Also of interest is that the length of the peptide chain provides a clue as to whether a transdermal transporter is involved. Specifically, dipeptides and tripeptides have been described as substrates for PepT1 binding and transport, which is a peptide transporter with a transmembrane electrochemical protein gradient located on the apical membrane of enterocytes [167]. In contrast, the TJs-mediated paracellular pathway is responsible for the translocation of oligopeptides containing four to nine amino acid residues [168,169]. This conclusion is corroborated by the work of the group of Xu et al. [42]. In an assay evaluating the uptake mechanism and bioavailability of rapeseed protein-extracted peptide WDHHAPQLR (RAP) in Caco-2 cell monolayers, they found that RAP is degraded by brush border peptidases, and then longer fragments of RAP, DHHAPQLR and WDHHAP are transported via the paracellular pathway, while tripeptide QLR is taken up via PepT1. In addition, through pharmacokinetic tests, they found that the absolute bioavailability of RAP (100 mg/kg BW) could reach 3.56% in rats after oral gavage, although the effective permeation rate of the basal side of Caco-2 monolayer measured in the preliminary screening test was only 1% at most, which was not sufficient to exert antioxidant effects. These results suggest that RAP may be developed as an efficient radical scavenging peptide. An earlier study by the same group showed that 65.57% of YWDHNNPQIR was resistant to hydrolysis by brush border peptidase and could be absorbed by the intestinal epithelium intact. More importantly, the absorbed peptides could still exhibit favorable antioxidant activity [170]. Wang et al. [82] reported an interesting work to screen and identify antioxidant peptides with digestive stability and bioaccessibility in muscle hydrolysates of silver carp. Two digestion products, i.e., viniferase and papain-induced hydrolysate fractions with molecular weight less than 1 kDa after SGID, showed 9.21% and 11.45% permeability by trans-cortical transport analysis of the Caco-2 monolayer. Then further identified by LC-MS/MS, the fragmented peptide LVPVAVF in the permeate showed the strongest DPPH^•^ cleavage (EC_50_ 0.65 mg/mL) and ROS quenching activity (27.23% at 50 µg/mL). Similarly, Feng and Betti [171] reported that digestion products of bovine collagen hydrolysate could reach up to 21.4% transport efficiency across Caco-2 cell monolayers. These studies support the idea that protein digests screened by in vitro permeability assays to obtain highly permeable fractions have greater potential as natural resistance components in food and drug systems than single or purified peptides. However, further studies focusing on the relationship between intestinal absorption of antioxidant peptides and subsequent in vivo metabolism are needed.

In conclusion, given the properties of antioxidant peptides, such as molecular weight, charge, hydrogen bond potential and hydrophobicity sensitivity to peptidase and intestinal transport, and the correlation between intestinal epithelial transport of peptides and peptidase catalysis, most oligopeptides exhibiting in vitro antioxidant activity rarely maintain their integrity or activity after transmembrane transport and subsequently affect their bioavailability, even though small amounts of target peptides may be detected in plasma. The presence of small amounts of the target peptide may be detected in plasma. As recently novel antioxidant peptides LHSMK [90], YFCLT and GLLLPH [169], WDHHAPQLR [42], GNPDIEHPE, SVIKPPTDE and VIKPPTDE [172] were reported as such. Therefore, to maximize health benefits, future work needs to shift more towards the development of promotion strategies for the stability and bioavailability of bioactive peptides.

## 5. Application of Antioxidant Peptides in Food Systems

Thanks to the natural properties and generally high biological activity of antioxidant protein hydrolysates or peptides, the use of these high value-added products as functional and nutritional fortification ingredients in specialty formulations is an area of increasing interest. Currently, the application of these active compounds is focused on four main areas: (1) as matrix enhancers, preservatives, or nutritional supplements in food systems; (2) as therapeutic agents to be incorporated into pharmaceutical systems; (3) as feed additives to improve animal immunity; and (4) development of anti-aging and photoprotective pharmaceuticals. Unlike living organisms, the quality features of food products suffer from irreversible decay from the date of production, which can be postponed from a few hours to months or even years if appropriate strategies are adopted. In this case, it turned out to be a common trend for the food industry to prioritize the use of synthetic antioxidants such as BHA (butylated hydroxyanisole), BHT (dibutylated hydroxytoluene), and TBHQ (tert-butylated hydroquinone) to promote food stability and extend shelf life [11]. However, given the potential health hazards of synthetic antioxidants, the current consumer trend is a dramatic increase in demand for ‘clean label’ foods and functional foods enriched with natural active ingredients. This trend has reinforced the demand of the food industry and researchers to obtain and apply food additives of natural origin that also exert bioactivity to prevent the development of NCDs. In this respect, antioxidant peptides have acted, at the laboratory level, as potential food additives. Few studies have evaluated the effects of antioxidant peptides in real food matrices, which could support their potential use as additives (Table 4). Meat products are a more studied food system as they are susceptible to lipid oxidation and require exogenous antioxidants to scavenge the active substances. For example, Shen et al. [173] reported that the addition of silver carp protein hydrolysate (2 and 4%, *w*/*w*) to surimi attenuated the formation of myofibrillar protein carbonyls, inhibited the reduction of free sulfhydryl content, and slowed down the formation of peroxidized lipid MDA and the rate of change of flavor compounds. Similarly, Lin et al. [174] proved that the incorporation of bighead carp gills hydrolysate (1 and 2%, *w*/*w*) treated with neutral protease to surimi increased the concentration of sulfhydryl and salt-soluble proteins, enhanced Ca^2+^-ATPase activity, reduced disulfide bonds, carbonyls, and hydrophobicity, and improved gel strength and texture. Nowadays, rarely commercial foods containing antioxidant active peptides exist on the market, despite the considerable literature on them. The possible reason is the lack of sufficient evidence for the biological effectiveness, processing and matrix stability, and toxicological safety of most antioxidant peptides. The stability of peptides during food processing and storage is critical for their application as functional ingredients, as peptides are vulnerable to chemical modifications of the backbone or side chains. These chemical reactions involve disulfide bond formation, dehydration, glycation, and aromatic ring oxidation, inducing changes in the structure and bioactivity of peptides [99,175]. The interaction between peptides and food matrix components such as proteins, lipids, and polysaccharides during food processing and storage could trigger a number of physicochemical reactions, such as hydrophobic interactions, disulfide interactions, and Maillard reaction, thereby favorably or adversely impacting the biological activity, solubility, sensory profiles, and color and texture parameters of peptide-based functional foods [99,176]. Over the decades, remarkable progress has been made in exploring the role of peptides in textural, sensory, and health aspects. Previous studies have focused on how peptides shape textural and technical functional properties, such as how mackerel gelatine hydrolysate affects flavor, color, emulsion activity and stability, and foaming properties and stability of carbonated beverage [86]. However, understanding how and to what extent peptides affect the functional properties of foods requires a comprehensive consideration of multiple complexities (e.g., peptide amphipathicity, solubility, and gelation capacity, food composition and ingredient distribution, and food processing and storage conditions) for better tailoring the type of hydrolysate in the formulated product to obtain the desired functional properties. Thus, further studies are requested to assess the impact of antioxidant bioactive peptides on the technical properties and consumer acceptance of final products, even though promising outcomes have been recorded in literatures (especially for minced meat and beverages).

## 6. Future Perspectives and Conclusions

As with other bioactive substances such as carotenoids, polyphenols, and flavonoids, the search and exploitation of protein hydrolysates or peptides with biological antioxidant activity has been guided by market demand since the beginning. Through the efforts of numerous researchers, a set of database-assisted bioinformatics evaluation methods has been established to complement the traditional proteomics approach, with the functions of identifying, characterizing, modifying, elucidating bioactive mechanisms, predicting structure-activity relationships, describing molecular interaction mechanisms, and producing bioactive peptides of food origin, which has contributed to the rapid development of bioactive peptide research. The rapid development of bioactive peptide research has been promoted. However, despite the benefits of antioxidant peptides, their commercialization faces multiple challenges. First, there is a lack of technically and economically feasible strategies for the preparation of bioactive peptides on an industrial scale. Current production methods are still relatively small-scale, time-consuming and expensive, and there is a lack of efficient and economical scale-up production techniques for the isolation and enrichment of peptide fractions. Second, the structure-activity mechanism of peptides has not been fully established. The clarification of this relationship can, on the one hand, help to directly predict the biological or chemical activities of peptide molecules by their quantitative structural characteristics (and vice versa), and on the other hand, provide a theoretical basis for peptide mutagenesis studies to design and construct peptides with specific biological functions or enhanced activities. Third, studies to establish the correlation between chemical and biological measurements are extremely limited. Further work is needed to explore in depth the possible links between chemical and biological measurements such as activity, digestive stability, and bioaccessibility. In addition, more in vivo investigations and clinical trials should be conducted here to provide the necessary biological support data for the health claims of active peptides. Fourth, as mentioned in Section 5, the activity and functional stability of peptides are easily disturbed by food matrix and food processing conditions. Moreover, their bitter taste, which cannot be easily removed, can reduce the sensory acceptability of fortified foods. Fifth, ensuring the stability and maximum bioavailability of peptides after ingestion is the most challenging part for researchers. However, many of the issues outlined previously can be addressed in a focused manner through nanoencapsulation. This is an economically and industrially effective delivery strategy for biofunctional components. In food systems, nanostructures can be used to protect peptides from a variety of adverse environmental conditions, enhance their water dispersibility, improve their matrix compatibility, and mask or reduce their unpleasant off-flavors. After ingestion, this means of delivery can optimize the release profile of peptides, improve its biostability, enhance its solubility in aqueous gastrointestinal solutions, and prolong its circulating half-life, thus allowing it to reach the target organ at an effective concentration. However, given the differences in bioactive properties, functional requirements, and the characteristics of food matrices and food processing, delivery systems must be carefully designed for each biopeptide’s practical application. Only a few studies have been conducted to introduce encapsulated biological components into food products. Last but not least, the hidden risk information of bioactive peptides, such as sensitization and toxicity, if any, must be fully uncovered in order to provide more complete evidence to support the market access of such substances.

In conclusion, this review provides a comprehensive summary of recent research advances on various food-derived antioxidant peptides. As a high value-added product with a sustainable source, the commercialization of antioxidant protein hydrolysates and purified peptides needs no further elaboration. With the rapid development of a range of emerging complementary production, analytical and bioavailability technologies, and their integration with multidisciplinary disciplines such as computer science, mathematical and statistical techniques, as well as animal and clinical medicine, the long-term goal of commercializing antioxidant peptides for edible products will be achieved.

## Figures and Tables

**Figure 1 biomolecules-12-01622-f001:**
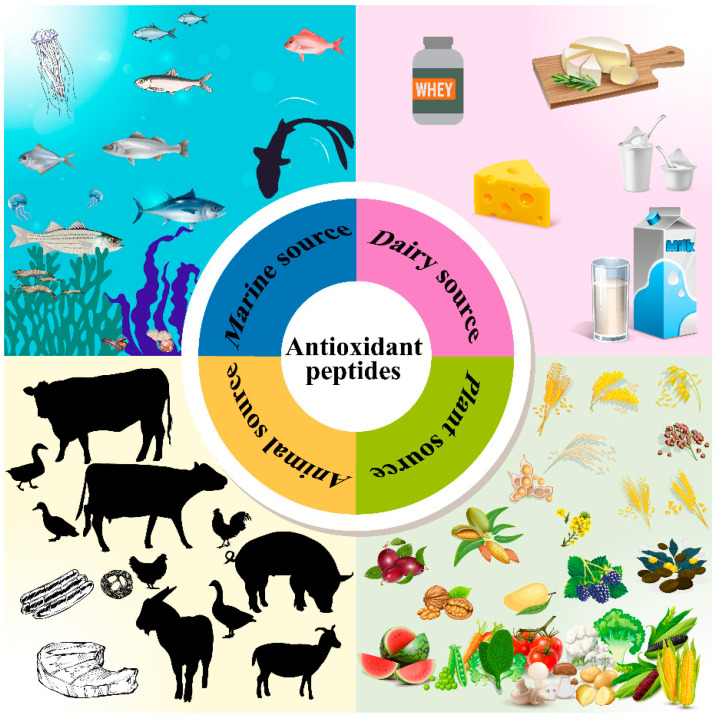
The main food-based sources of antioxidant active peptides.

**Figure 2 biomolecules-12-01622-f002:**
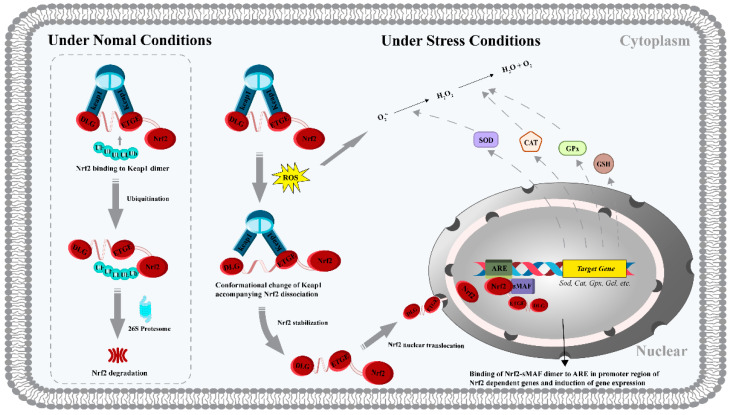
Schematic showing the regulatory mechanism of the Keap1-Nrf2-ARE pathway. Under normal physiological conditions, the Keap1 dimer binds to two binding motifs, called DLG and ETGE, in a specific domain of Nrf2, allowing Nrf2 to be sequestered in the cell membrane and maintained at low levels by Keap1-dependent ubiquitination and proteasomal degradation. Under oxidative stress, such as the presence of reactive oxygen species (ROS), the conformation of Keap1 changes, resulting in the release of Nrf2 from Keap1-directed degradation, which translocates in the nucleus and forms a dimer in sMAF. The polymer formed leads to the induction of ARE-dependent genes such as Sod, Cat, Gpx, and Gcl. These gene products subsequently exert cytoprotection against ROS. Ub, ubiquitin.

**Figure 3 biomolecules-12-01622-f003:**
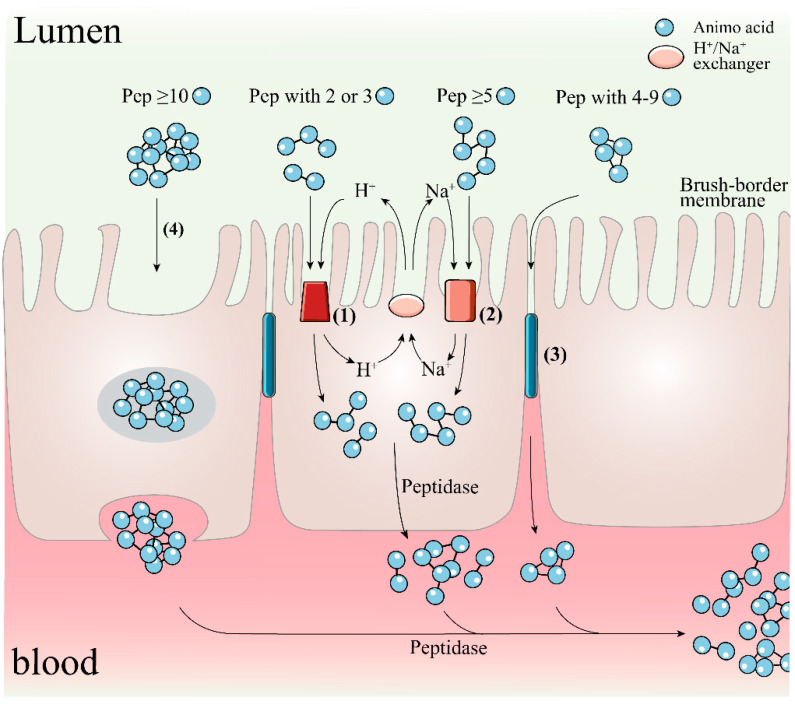
Underlying mechanisms of transcortical transport of peptides with diverse sizes. (1) H+-coupled PepT1 and 2; (2) Na+-coupled SOPT1 and 2; (3) paracellular; (4) transcytosis. Pep, peptide.

**Table 4 biomolecules-12-01622-t004:** the use of antioxidant protein hydrolysates (peptides) in food systems.

Source	Extraction Tool	Hydrolysate/ Peptide	Product	Effects	Ref.
Silver carp	Protamex	Silver carp protein hydrolysate (2 and 4%, *w*/*w*)	Surimi	Delayed the formation of MDA and unfavorable flavor volatiles; inhibited the oxidation of free sulfhydryl and the formation of carbonyls in myofibrillar proteins	[173]
Bovine	Pepsin	TSKYR (0.1 and 0.5%, *w*/*w*)	Ground beef	0.5% TSKYR provided similar protection against lipid oxidation as 0.1 and 0.5% BHT	[177]
Bighead carp	Flavourzyme Alcalase Neutral protease Papain	Bighead carp gill protein hydrolysate (1 and 2%, *w*/*w*)	Surimi	Increased sulfhydryl and salt-soluble protein concentrations, enhanced Ca^2+^-ATPase activity, reduced disulfide bonds, carbonyls and hydrophobicity, and improved gel strength and texture	[174]
Barred mackerel	Alcalase and actinidin	Barred mackerel gelatine hydrolysate (<3 kDa)	Carbonated beverage	No adverse effects on emulsification activity and stability, foam expansion and stability, color, and flavor	[86]
Faba bean seed	Pepsin Trypsin Alcalase	Faba bean hydrolysate (1%, *w*/*v*)	Apple juice	No adverse effects on organoleptic acceptability	[9]
Capelin	Alcalase Neutrase Papain	Capelin protein hydrolysate (0.5–3.0%, *w*/*w*)	Ground pork	Inhibited 17.7–60.4% of TBARS production; increased cooking yield (up to 4% at 3.0%)	[178]
Goby	Grey triggerfish proteases	Goby protein hydrolysate (0.01–0.2%, *w*/*w*)	Turkey meat sausage	0.02% and 0.04% hydrolysate showed higher MDA inhibition capacity than 0.2% vitamin C	[179]

## Data Availability

The data presented in this study are available on request from the corresponding authors.

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
