# Peer review of "Food Protein-Derived Antioxidant Peptides: Molecular Mechanism, Stability and Bioavailability"

_biomolecules, 2022, doi:10.3390/biom12111622_

Round 1
Reviewer 1 Report
The review "Advances on the antioxidant peptides from edible food resources" is interesting and comprehensive but certain points required more work by the authors.
Among these points:
1- The authors should modify the abstract to mention the methods used to collect data, how many references and how many peptides were cited and many other details that summarize the key findings
2- Tables for summarizing the Methods of activity evaluation and mechanisms of action of antioxidant peptides will be very helpful
3- The authors should give their comments on the citing data not only citing the knowledge by copy and paste the paraphrasing
4- The authors should enhance the application in food systems which is highly required
5- The manuscript should be checked by an English native speaker to remove some syntax
Author Response
Thank you sincerely for your review. Please see the attachment.

Reviewer 2 Report
Thank you for the opportunity to review this work. The paper is excellent. I congratulate the authors and I will use the content, as soon as it is published, in our research group.
Author Response
Thank you sincerely for your review.
Reviewer 3 Report
Dear authors,
Attached you can find a document with comments and suggestions that would contribute to an even better quality of your review paper.
Note: The reference list should include the full title, as recommended by the ACS style guide, and abbreviated names of first author's names and journal's names. Please format the reference list in accordance with the journal's guidelines.
Best regards.

Author Response

(The authors gave the same response as above.)

Round 2
Reviewer 1 Report
The manuscript has much improved however, points 3 and 4 are not addressed well
the authors can read the following review to get and idea about how to correlate his work with the application in food industry and how to discuss and write future perspectives
Jia et al. 2021 Bioactive peptides from foods: production, function, and application Food & function
Author Response
Thank you sincerely for your valuable comments on the article. Please see the attachment for details of the revisions.
